# Indoxyl Sulfate Stimulates Angiogenesis by Regulating Reactive Oxygen Species Production via CYP1B1

**DOI:** 10.3390/toxins11080454

**Published:** 2019-08-02

**Authors:** Jiayi Pei, Rio Juni, Magdalena Harakalova, Dirk J. Duncker, Folkert W. Asselbergs, Pieter Koolwijk, Victor van Hinsbergh, Marianne C. Verhaar, Michal Mokry, Caroline Cheng

**Affiliations:** 1Department of Nephrology and Hypertension, DIGD, UMC Utrecht, University of Utrecht, 3584 CX Utrecht, The Netherlands; 2Department of Cardiology, Division Heart & Lungs, UMC Utrecht, University of Utrecht, 3584 CX Utrecht, The Netherlands; 3Regenerative Medicine Utrecht, UMC Utrecht, University of Utrecht, 3584 CX Utrecht, The Netherlands; 4Department of Physiology, Amsterdam UMC, VUmc location, Amsterdam Cardiovascular Science, 1081 HV Amsterdam, The Netherlands; 5Department of Pathology, UMC Utrecht, University of Utrecht, 3584 CX Utrecht, The Netherlands; 6Department of Cardiology, Erasmus MC, 3015 GD Rotterdam, The Netherlands; 7Institute of Cardiovascular Science, Faculty of Population Health Sciences, University College London, London NW1 2DA, UK; 8Health Data Research UK and Institute of Health Informatics, University College London, London NW1 2DA, UK; 9Department of Clinical Chemistry and Heamatology, University of Utrecht, 3584 CX Utrecht, The Netherlands; 10Division of Paediatrics, UMC Utrecht, University of Utrecht, 3584 CX Utrecht, The Netherlands

**Keywords:** indoxyl sulfate, chronic kidney disease, reactive oxygen species, CYP1B1, angiogenesis

## Abstract

Indoxyl sulfate (IS) is an accumulative protein-bound uremic toxin found in patients with kidney disease. It is reported that IS impairs the vascular endothelium, but a comprehensive overview of all mechanisms active in IS-injury currently remains lacking. Here we performed RNA sequencing in human umbilical vein endothelial cells (HUVECs) after IS or control medium treatment and identified 1293 genes that were affected in a IS-induced response. Gene enrichment analysis highlighted pathways involved in altered vascular formation and cell metabolism. We confirmed these transcriptome profiles at the functional level by demonstrating decreased viability and increased cell senescence in response to IS treatment. In line with the additional pathways highlighted by the transcriptome analysis, we further could demonstrate that IS exposure of HUVECs promoted tubule formation as shown by the increase in total tubule length in a 3D HUVECs/pericytes co-culture assay. Notably, the pro-angiogenic response of IS and increased ROS production were abolished when *CYP1B1*, one of the main target genes that was highly upregulated by IS, was silenced. This observation indicates IS-induced ROS in endothelial cells is *CYP1B1*-dependent. Taken together, our findings demonstrate that IS promotes angiogenesis and *CYP1B1* is an important factor in IS-activated angiogenic response.

## 1. Introduction

Indoxyl sulfate (IS) is a uremic retention solute that accumulates in the systemic circulation due to renal impairment [1]. Unlike other uremic retention solutes that are water-soluble or non-protein-bound, IS binds to albumin (66.5 kDA) and cannot be cleared effectively via dialysis, which is the main method to remove uremic toxins in end-stage chronic kidney disease (CKD) patients [1]. Compared to healthy individuals, serum IS levels are nearly 50 times higher in patients with acute kidney injury (AKI) and reach the highest level in patients with end-stage CKD [2]. Circulating IS has been shown to play an important role in the progression of CKD and the development of cardiac disease, such as left ventricular hypertrophy [3]. Although treatment with the carbonaceous adsorbent AST-120 to lower serum IS level showed improvement on renal and cardiac function in both animal models and a phase II study, it failed to demonstrate promising results in the subsequent phase III trail [4,5]. More studies based on a genome wide analysis approach could shed new light on the working mechanism of IS associated cardiorenal disease and provide new targets for the development of new therapeutic approaches.

Studies have implied that IS first impairs endothelial function, which subsequently contributes to the worsening of kidney function and the development of cardiovascular disease [6]. Flow-mediated endothelial dilation (FMD), which is a clinical parameter for endothelial function, is significantly lower in CKD patients when compared to healthy individuals and hypertensive patients [7]. Notably, by lowering IS level, FMD increases in CKD patients and correlates inversely with IS levels. In vitro studies have revealed that IS inhibits nitric oxide (NO) production, which is a critical regulator of vascular tone, while it promotes reactive oxygen species (ROS) release, resulting in oxidative stress [7,8,9]. Furthermore, IS inhibits the proliferation ability of endothelial cells (ECs) by activating aryl hydrocarbon receptor-mediated cell senescence [10]. Besides the direct deleterious effect of IS on ECs, IS also interferes with the immune system and actives inflammatory cytokines, such as IL-1β, E-selectin and TNF-α, which contribute to EC apoptosis and result in endothelial dysfunction [11,12]. Thus IS appears to have a broad and complex effect on endothelial function. A genome-wide transcriptome study would aid in mapping the key driving signalling factor(s) underlying this important IS-induced disease mechanism.

In the present study, we conducted a genome-wide transcriptome analysis using RNA sequencing (RNA-seq) to reveal the transcriptome profile of IS-treated human umbilical vein endothelial cells (HUVECs). Based on the set of IS influenced genes, we performed gene enrichment analyses and obtained indications on altered pathways involved in cell migration, angiogenesis, apoptosis and cell metabolism. We studied these enriched biological processes at the functional level using an established 3D collagen-based in vitro model for angiogenesis [13,14]. An activated angiogenic response was observed under IS stimulation. Furthermore, cytochrome P450 1B1 (*CYP1B1*) was identified as one of the strongest up-regulated genes in IS-treated HUVECs. Silencing of *CYP1B1* decreased IS-induced ROS and attenuated angiogenic response under IS stimulation, implying a role for *CYP1B1*-dependent ROS production in the IS-induced angiogenic response.

## 2. Results

### 2.1. RNA-seq Reveals Differentially Expressed Genes in IS Treated HUVECs

We studied transcriptome changes in HUVECs after 24 h stimulation of 250 µM IS as compared to the potassium salt (KCl) control using RNA-seq. Heatmap depicts clustering of samples based on all differentially expressed genes between the two groups (Figure 1A). Volcano plot shows both fold change and *p* value of all genes in log 2 scale, and differentially expressed genes were highlighted in red (Figure 1B). In total, we identified 1293 genes that were differentially expressed between KCl control and IS groups (*p* value < 0.05, Table 1, Appendix A). Of these, 643 genes were up-regulated by IS as compared to the control group, and gene enrichment analysis showed that they were mostly involved in cell migration, angiogenesis and programmed cell death processes (Figure 1C); 650 genes were down-regulated by IS, and these were mainly enriched for biological processes related to cell metabolism, such as cell cycle process, chromosome segregation, and cell division (Figure 1D).

### 2.2. IS Inhibits Cell Viability at High Concentration

As the transcriptome profile indicated that apoptosis was enhanced by IS in HUVECs, we performed a MTT assay to examine the viability of IS-treated HUVECs. IS at 250 µM decreased cell viability as compared to the control, although the decrease was not significant (*p* value = 0.07, Figure 2A). A significant decrease of cell viability was achieved in 500 µM and 750 µM IS treated HUVECs (*p* value < 0.05, Figure 2A).

### 2.3. IS Induces Cell Senescence

The transcriptome data also indicated that cell cycle progression was impeded by IS exposure. For validation, we examined IS-induced senescence features. HUVECs were incubated with 250 µM IS or control buffer for 24 h incubation before the X-gal assay for senescent cell detection. We observed more X-gal positive cells (blue) in IS group when compared to the control (Figure 2B). In line with this finding, the expression level of cell senescence marker *CDKN1A* was significantly higher in IS group when compared to the KCl control, whereas the expression level of cell proliferation marker *KI67* was significantly lower in IS group, as shown by RT-qPCR validation (Figure 2C).

### 2.4. IS Does Not Influence Cell Migration Ability

Transcriptome data implied that IS enriched transcripts of genes were involved in cell migration. We used both invasive wound healing assay and non-invasive plug assay to study the influence of IS on the cell migration capacity. After 24 h incubation with either IS or KCl control buffer, the migration distances of HUVECs were comparable between two groups at three difference concentrations using the wound healing assay (Figure 2D,E). Likewise, no difference was detected between the two groups on the number of migrated HUVECs into the cell-free area using the plug assay (Figure 2F,G).

### 2.5. IS Promotes Angiogenic Response

The main biological process enriched in upregulated genes of the IS response was blood vessel morphogenesis. To validate this, we used a 3D collagen-based co-culture model to study the influence of IS on endothelial reorganization and tubule formation. In this assay, HUVECs-GFP cells and pericytes-DsRED were cultured together in type I collagen and incubated with IS or control buffer for 3 days. In control conditions, these vascular cells will undergo EC sprouting, tubule formation and pericytes-induced stabilization of neovascular structures in 3–5 days. Compared to the standard culture medium, the KCl-adjusted control medium did not affect formation of vascular structures. No difference on the number of branches, the number of tubules and total tubule length was detected between IS and KCl control treated co-cultures at day 1 post-stimulation. At day 3 both the number of branches and tubules in IS groups were higher than the controls (*p* value = 0.063 and *p* value = 0.062 respectively). The total tubule length in IS group was significantly higher compared to control (*p* value < 0.05, Figure 3A,B). Combined, these in vitro data, except for the migration assays, confirm the findings from transcriptome analysis, and demonstrate the complex effects of IS on ECs homeostasis and regenerative capacity.

### 2.6. Depletion of CYP1B1 Inhibits Tubular Formation

*CYP1B1* showed the highest fold change among IS up-regulated genes based on the RNA-seq data. To investigate the role of *CYP1B1* in IS-treated HUVECs, we first validated its expression level using RT-qPCR and obtained a consistent result (Figure 4A). Compared to the siSham transfected HUVECs, a significant lower mRNA expression level of *CYP1B1* was observed in *CYP1B1* silenced HUVECs 3 days post transfection (Figure 4B). Next, we investigated the possible involvement of CYP1B1 in relation to IS-induced angiogenesis in the previously described co-culture assay. After exposure to 250 µM IS, the number of branches, the number of tubules and total tubule length were significantly lower in *CYP1B1* silenced HUVECs when compared to siSham transfected cells at day 3, and remained highly suppressed at day 4 after IS stimulation (Figure 4C,D).

### 2.7. CYP1B1 Plays an Important Role in IS-Increased ROS Production

ROS plays an important role in the induction of endothelial dysfunction and has been shown to trigger the angiogenic response [15]. Next, we studied the possible involvement of *CYP1B1* in IS-induced ROS production. In line with previous reports, IS-treated HUVECs showed significantly higher ROS level when compared to the control at three different concentrations (*p* value < 0.05, Figure 5A,B), indicative of enhanced cellular oxidative stress. However, HUVECs transfected with siRNA targeting *CYP1B1* transcripts demonstrated a decrease in ROS production compared to control sisham transfected groups at three difference concentrations (Figure 5C,D). Additionally, we further investigate this effect in cardiac microvascular endothelial cells (CMECs) to access whether it holds true in the arterial vascular bed, in particular cardiac microcirculation. Notably, we also observed a significantly increase of ROS production in IS-treated CMECs, which was attenuated after silencing *CYP1B1* (Figure 5E). Combined, these data indicate a regulatory role of *CYP1B1* in endothelial ROS production.

## 3. Discussion

In the present study, we demonstrated using whole genome transcriptome analysis that the IS affected genes in ECs were mostly enriched in biological functions related to vascular formation, cell apoptosis, and cell cycle. Consistent with previous studies, we validated in in vitro assays that IS indeed induced an EC phenotype with reduced cell viability and increased activated cellular senescence [9,10]. Paradoxically, we also observed enhanced angiogenic capacity of vascular cells in our 3D co-culture system in response to IS stimulation, which was in line with our transcriptome findings. We identified CYP1B1 as a new downstream target of IS and demonstrated the pro-angiogenic effect of IS was likely to be regulated via CYP1B1 modulation of endothelial ROS levels.

Dou and colleagues showed that IS decreased cell proliferation ability of HUVECs, but it did not affect cell viability at tested concentrations (from 100 μM to 1 nM) using the trypan blue exclusion test, which stains only dead cells [16]. We did not observe an effect on cell viability after 250 μM IS stimulation, however we showed that 500 μM and 750 μM impaired cell viability using the MTT assay, which more reflects metabolic activity. Dou et al. also showed that in the presence of 4% human albumin, IS decreased the wound repaired at the concentration of 125 μg/mL and 250 μg/mL. This reduction remained but was milder in IS-treated cells without the addition of albumin. On the contrary, we did not find any effect of IS on cell migration and would healing ability, which might be explained by differences in assay setups: no albumin was used in our stimulation buffer and the effect we obtained might resemble more the non-bound IS. However, the exact effect of albumin binding of IS on endothelial cell response remains to be further evaluated.

Patients with AKI suffer from oxidative stress, during which oxygen radicals could lead to cell injury and trigger apoptosis and senescence [17,18]. The application of antioxidants in lowering ROS level and to modulate AKI has been extensively studied and reviewed in previous studies [19]. In CKD patients, ROS level remains at a high level, especially in patients with end stage kidney disease, and has been proposed as an important mediator in CKD-associated cardiovascular diseases [20,21]. Consistent with previous studies showing the ability of IS to induce ROS [6,9], we also showed IS-induced ROS production. IS is a protein-bound toxin and around 90% of IS bind to plasma proteins [22,23]. Most previous in vitro studies used a range from 62.5 μM to 1000 μM “free” IS [24,25,26] or 2 mM to 20 mM protein-bound IS [27,28]. Additionally, 250 μM IS is comparable to the mean serum level in CKD patients [29,30], and the maximum IS concentration in the circulation of patients is approximately 236 mg/L (939.1 μM) as reported by the European Uremic Toxin Work Group [31]. Therefore, we examined IS at a broad range of concentrations from 250 μM to maximum 750 μM. Furthermore, we observed this effect in both HUVECs and CMECs, suggesting a potential role of IS-induced ROS in cardiorenal syndrome.

Excessive ROS has been shown to promote angiogenesis by inducing proangiogenic factors in ECs, such as VEGF, MMPs, ANGPT1, and VEGFRs [32]. ROS also oxidize phospholipids and the resulting oxidant products could contribute to angiogenesis via TLR signalling [15]. The transcriptome data indicated that IS exposure significantly increased expression levels of VEGFC, MMP1, MMP24-AS1 and MMP25-AS1 in ECs. Furthermore, in our co-culture assay, we found a significant increase in total tubule length 3 days after exposing to IS, indicating IS activated a (micro)vascular angiogenic response.

A major source of ROS is cytochrome P450 activity [33]. Cytochrome P450 is a large complex of enzymes, which are actively involved in more than 70% of all drug metabolism by initiating monooxygenase or hydroxylation reaction via other substrates (i.e., oxygen and NADPH) [34]. During the reaction, P450 produces active oxygen species and subsequently contribute to excessive ROS formation [33].

CYP1B1, the biggest known human P450 protein in terms of size of mRNA and amino acids, is highly expressed in tumour cells and studies have highlighted its important role in tumour development [35]. Compared to the general population, the prevalence of cancer is higher in patients with moderate CKD and patients received dialysis or kidney transplantation [36]. Notably, McFadyen and colleagues showed a higher CYP1B1 expression in renal cell carcinoma when compared to the normal kidney [37]. Gondouin et al. further showed that IS increased CYP1B1 expression in HUVECs using a microarray setup [38]. Consistent with previous studies, we also demonstrated an increased in CYP1B1 expression levels in response to IS. In fact, our RNAseq based analysis showed that *CYP1B1* had the highest fold change increase among all IS-activated genes. So far, only a limited number of studies have shown the involvement of CYP1B1 in angiogenesis. Dallaglio and colleagues showed that both the RNA and protein expression levels of CYP1B1 were significantly down-regulated in HUVECs after exposing to metformin, which inhibited vascular formation in vitro [39]. Tang and colleagues showed that the number of retinal blood vessels was decreased in mice that lacked CYP1B1 [40]. They also confirmed that the lack of CYP1B1 impaired endothelial cell sprouting in vitro, which could be reversed by restoring CYP1B1 expression. Both studies used only one vascular cell type (ECs) in a 2D Matrigel-based model. A later study from Palenski and colleagues examined the involvement of CYP1B1 in both retinal ECs and pericytes that were isolated from mice [41]. They also observed impaired vascular formation using the Matrigel model, in which ECs that lacked CYP1B1 were cultured with normal pericytes. Expanding on these previous findings, using our established 3D type I collagen human-derived EC and pericyte co-culture model that allows complex vascular structure formation [13,14], we provide evidence that the enhanced angiogenic response under IS stimulation is partially mediated via CYP1B1 upregulation by IS in endothelial cells.

Interestingly, multiple studies show defective angiogenesis in CKD patients. Futrakul and colleagues showed that CKD patients had nearly 17-fold increase of circulating endothelial cells that reflected vascular injury when compared to healthy individuals. This was also linked to a decrease in VEGF/endostatin ratio that indicated a decline in angiogenic capacity [42]. A recent study included a larger population of both CKD patients and healthy individuals, and showed a decrease in angiopoietin-1/VEGF-A ratio in CKD patients when compared to the control, indicating impaired angiogenesis and enhanced endothelial leakage [43]. In AKI, hypoxia impaired angiogenesis has also been identified, which has been proposed to contribute to the transition from AKI to CKD [44]. However, only a limited number of studies are focused on the influence of IS on angiogenesis. Hung and colleagues showed that accumulated IS in nephrectomised mice inhibited the maturation of endothelial progenitor cells (EPCs) and subsequently suppressed neovascularization [45]. By treating these mice with AST-120 that removes IS precursor indole in the intestine, they showed a decreased plasma level of IS and an improvement in the EPC-based neovascularization. Another study showed IS inhibited the chemotactic motility and the colony-forming ability of human EPCs [46]. In our study, IS activated genes were highly enriched for angiogenesis. We also showed increased branches and tubule formation at 3 days after IS stimulation in vitro, and the total tubule length was significantly higher in IS group. Combined, our data suggest enhanced vascular formation activity in response to IS. It is important to point out that different cell types were used in previous studies and in our study, namely EPCs and HUVECs respectively [47]. Besides, instead of vasculogenesis during which vascular formation occurs from in situ differentiating EPCs [48], we used a well-established model to study angiogenesis of differentiated ECs. Additionally, tubular structures seem to decline from day 3 post IS stimulation to day 4 in *CYP1B1* silenced cells, whereas it remained stable or slightly increased between day 3 and day 4 in absence of IS stimulation (Appendix A). Further studies are required to examine the long-term effect of IS on angiogenesis. Combined, they could explain the different findings between previous studies and the present study.

IS has been shown to function as a ligand for aryl hydrocarbon receptor (AhR), and the AhR signaling is activated upon binding [49]. Notably, the expression level of AhR is relatively high in kidney [50], and the highest level of accumulated IS has also found in kidney as compared to lung, heart and liver [51]. A positive correlation between the increased activation of AhR signaling and CKD has also been shown [52], suggesting the deleterious effect of AhR signaling in kidney disease. The downstream targets of AhR signaling vary among cell types [53,54,55,56]. To investigate the regulation between IS and AhR signaling in ECs, we collected 266 established AhR targets (systematic name: M9986 and M17378) from Molecular Signature Database v6.2 and performed gene set enrichment analyses (GSEA) to examine the representation level of AhR targeted genes in differentially expressed genes from our study. We observed an over-representation of AhR downstream targets in the IS-activated genes under the default settings (FDR < 25%, Appendix A and Appendix A), implying the activation of AhR pathway by IS in HUVECs. Taken together, these data highlight an important main regulatory pathway through which IS could negatively impact the regenerative capacity of the renal vasculature in renal disease. More studies are required to investigate the effect of IS-mediated AhR signaling on angiogenic response in ECs.

In terms of disease, we previously reported increased capillary networks in heart and kidney of obese ZSF1 rat with cardiorenal metabolic syndrome when compared to the controls, indicating an activated angiogenic response [57]. Despite the enriched ECs foci and pericytes foci, the lack of regular vascular endothelial luminal surface and the decrease of peritubular and glomerular endothelium suggested non-functional vasculature. Furthermore, they recruited macrophages, which subsequently contributed to the fibrotic formation. It is important to note that other risk factors, such as hypertension and onset of heart failure with preserved ejection fraction, were also observed in this obese ZSF1 rat model and could interfere with the impaired vascular formation. In the light of these findings, the observed IS-activated angiogenic responses need further investigation, especially in relation to their functional activities, the paracrine signaling with the immune system and the possible influence from other metabolic risk factors.

In summary, we presented a comprehensive list of IS affected genes in ECs. Gene enrichment analyses indicated altered angiogenesis and cell metabolism. IS induced enhanced ROS production in ECs, which was CYP1B1-dependent. Furthermore, IS activated an angiogenic response in HUVECs-pericytes co-culture. CYP1B1 deficiency in ECs resulted in a suppressed angiogenic response, indicating a critical role of CYP1B1 in IS-activated angiogenesis. We hypothesize that IS induces ROS level in ECs, which initiates the activation of the observed angiogenic responses. However, the influence of a chronic status of high ROS level on the balance between pro- and anti-angiogenic factors in vivo is more complex and remains to be elucidated. With the transcriptome data generated in our study, we offer a detailed overview of putative functional chances in EC-behaviour in response to IS, and we identified CYP1B1 as a key regulator in the process, shedding light into the underlying mechanism of IS-regulated vascular formation and maturation.

## 4. Materials and Methods

### 4.1. Cell Culture

Human umbilical vein endothelial cells (HUVECs) were cultured in EGM2 medium (Lonza, Breda, The Netherlands) with 100 UmL^−1^ penicillin-streptomycin (PS). Pooled donor HUVEC were purchased from Lonza and used in all functional assays in this study. For the sequencing purpose, HUVECs were isolated from three newborns anonymously and obtained from the University of Utrecht Department of Gynecology (The Netherlands), with the informed consent under the EPD term. Human brain vascular pericytes (ScienCell, Uden, The Netherlands) were cultured in DMEM (Gibco, Landsmeer, The Netherlands) supplied with 10% FCS and 100 UmL^−1^ PS. All cells were cultured in the gelatin-coated dishes (Greiner Bio-One, Alphen aan den Rijn, The Netherlands) in a 5% CO_2_ incubator at 37 °C. Cells between passage 3 and 8 were used in this study.

### 4.2. RNA-seq and Data Analysis

HUVECs were incubated with 250 µM IS for 24 h. Potassium chloride (KCl) was used as a control, because IS is a potassium salt. Total RNA isolation was isolated using the RNeasy Mini Kit (Qiagen, Venlo, The Netherlands) according to the manufacturer’s recommendations. Polyadenylated mRNA was further selected using Poly(A) Beads (NEXTflex^TM^) and libraries were generated using the NEXTflex^TM^ Rapid RNA-seq Kit (Bioo Scientific, Uden, The Netherlands). Libraries were sequenced by the Nextseq500 platform (Illumina, San Diego, CA, USA). Sequencing data were analysed as described previously [58]. Briefly, reads were aligned to the human reference genome GRCh37 and mapped to the transcriptome. Reads per kilobase million for each refseq gene were calculated [59] and a list of differentially expressed genes between IS and control groups was obtained at *p* value < 0.05 [60].

### 4.3. Gene Enrichment Analysis

Differentially expressed genes were enriched for their biological functions using ToppGene Suite tool ToppFun (default setting: FDR correction, *p* value cut off at 0.05 and gene limit set between and including 1 and 2000 per pathway) [61].

### 4.4. Cell Metabolism Assay

Cells were seeded to a gelatin-coated 96-well plate and incubated overnight for adhesion. Cells were washed once with PBS and incubated with either IS or control buffer for 24 h. Afterwards, stimulation buffers were removed and cells were washed once with PBS, followed by 4 h incubation with 100 µL 3-(4,5-dimethylthiazol-2-yl)-2,5-diphenyltetrazolium bromide (MTT) buffer (0.5 mg/mL, Sigma, Zwijndrecht, The Netherlands). MTT buffer was removed and 200 µl dimethyl sulfoxide (DMSO) per well was added to dissolve formed formazan crystals. The plate was shaken gently using a microplate shaker (IKA, Staufen, Germany) for 30 min in the dark. Absorbance was measured at 570 nm by the microplate reader (Bio-Rad, Veenendaal, The Netherlands). To correct for batch effects, an additional condition of cells cultured in the standard growth medium was included in each independent experiment.

### 4.5. Senescence-Associated Beta Galactosidase Activity

Cellular senescence was examined using the Senescence Detection Kit (Abcam, Cambridge, UK). Briefly, treated HUVECs were washed once with PBS and fixed using Fixative Solution for 15 min in the incubator. Afterwards, HUVECs were washed twice with PBS and incubated with staining buffer containing 25 mg/mL X-gal overnight. TNFα (40 ng/mL) treated HUVECs were used as positive control. Images of random views were taken at 10× magnification using an inverted fluorescence microscope.

### 4.6. Wound Healing Assay

Cells were seeded to a gelatin-coated 24-well plate (Greiner Bio-One) and were grown till 95% confluency in the growth medium. A scratch was made to create a cell-free area. Cells were washed once with PBS and incubated with IS or control buffer. Images were taken at 0 h and 24 h post stimulation. Area covered by migrated cells from the leading edge of the scratch was measured. To avoid batch effect, an additional condition of cells cultured in the standard growth medium was included in each independent experiment.

### 4.7. Cell Migration Plug Assay

Cell stoppers (Oris^TM^) were pre-inserted to a gelatin-coated 96-well plate (Greiner Bio-One) to create a cell-free area. Cells were seeded to the plate and incubated overnight for adhesion. The following day cell stoppers were removed and cells were stimulated by IS or control buffer for 24 h. Afterwards, cells were stained by calcein-AM (BD-Bioscience, San Jose, CA, USA) and images were taken by.

### 4.8. 3D Collagen Co-Culture Assay

Lentivirus green fluorescent protein transduced HUVECs (HUVECs-GFP) and lentivirus discosoma sp. Red fluorescent protein transduced pericytes (pericytes-DsRED) were mixed at 5:1 ratio in co-culture medium, which is basal EBM medium supplied with 2% FCS, rhFGF-B, ascorbic acid and 100 UmL^-1^ PS. Cell mixture supplied with growth factors, including IL-3, SCF-1 and CXCL12 (BD Bioscience), at the volume of 300 µL was added to 200 µL bovine collagen type I (Gibco). NaOH was used to adjust pH to 7.5. Cell-collagen mixture was added to 96-well plate (50 µL per well). After 1 h incubation, 100 µL EGM2 was added per well and the plate was incubated overnight. The following day IS or control buffer was added to the cells. Images were taken from day 1 till day 3 after stimulation using inverted fluorescence microscope and analysed by AngioSys 2.0. To correct for batch effects, an additional condition of cells cultured in the standard co-culture medium was included in each independent experiment.

### 4.9. Detection of Intracellular ROS Levels in HUVECs

Cells were seeded to a gelatin-coated 96-well plate and incubated overnight for adhesion. Next day, IS or control buffer was added to the cells. After 24 h, stimulation buffer was removed and cells were exposed to 10 µM CM-H2DCFDA (Life Technologies, Landsmeer, The Netherlands) for 30 min in the dark. Excessive CM-H2DCFDA was washed twice with PBS, followed by the addition of 100 µL PBS supplied with 0.2% Bovine Serum Albumin. Images were taken using a SP8X confocal microscope (Leica, Amsterdam, The Netherlands) at 20× magnification, and the fluorescence intensity was measured at the wavelength of 485 nm (excitation) and 538 nm (emission) using a fluorescence plate reader (Fluoroskan^TM^).

### 4.10. Detection of Intracellular ROS Levels in Cardiac Microvascular Endothelial Cells (CMECs)

Besides HUVECs, we also examined the impact of IS on ROS production in cardiac microvascular endothelial cells (CMECs). The culture of CMECs (Lonza, CC-7030) and cytoplasmic reactive oxygen species measurement were performed as previously described [62]. Briefly, 6 h after exposure to 250 μm IS, CEMCs were washed once and incubated with 5 µM CM-H2DCFDA (C6827, ThermoFisher, Landsmeer, The Netherlands) for 30 min in phosphate buffer saline (220/12257974/1110, Braun, Landsmeer, The Netherlands). Excess CM-H2DCFDA was washed off. Images were taken using a Zeiss Axiovert 200M Marianas inverted fluorescence microscope (Intelligent Imaging Innovations, Denver, CO, USA) with a 63× oil-immersion objective at 37 °C and 5% CO_2_ environment. All fluorescent images were corrected for background and negative controls. Quantification of all fluorescent images was performed using digital cell masking software (Slidebook 6, Intelligent Imaging Innovations).

### 4.11. Reverse Transcription-Quantitative Polymerase Chain Reaction (RT-qPCR) Analysis

RNA was isolated from HUVECs treated by 250 µM IS for 24 h or control buffer. The quality and quantity of RNA was measured by a spectrophotometer (DeNoVIX, Waddinxveen, Landsmeer, The Netherlands). Complementary DNA was transcribed using iScript Synthesis Kit (Bio-Rad) according to the manufacturer’s instructions. RT-qPCR was performed to measure mRNA level of targeted genes using SYBR-GREEN-Cycler IQ5 detection system (Bio-Rad). β-actin was used as the housekeeping gene. Primer sequences were as follows: *CYP1B1* (forward: 3′-TGATGGACGCCTTTATCCTC-5′; reverse: 5′-ACGACCTGATCCAATTCTGC-3′), *CDKN1A* (forward: 3′-GACACCACTGGAGGGT GACT-5′; reverse: 5′-ACAGGTCCACATGGTCTTCC-3′), *KI67* (forward: 3′-AAGCCCTCCAGCTC CTAGTC-5′; reverse: 5′-TCCGAAGCACCACTTCTTCT-3′), and beta-actin (forward: 3′-TCCCTG GAGAAGAGCTACGA-5′; reverse: 5′-AGCACTGTGTTGGCGTACAG-3′).

### 4.12. Short Interference RNA

HUVECs were grown to 60% confluence and transfected with 200 nM *CYP1B1*-siRNA or non-targeting Sham-siRNA (Dharmacon^TM^, Athens, Greece) using lipofectamine according to the manufacturer’s instructions. The silencing effect of *CYP1B1* expression in HUVECs was validated using RT-qPCR at 24 h and 72 h post transfection. Successfully transfected cells were further used for comparing the intracellular ROS production and angiogenic response between IS and control groups as explained above.

### 4.13. Statistical Analyses

Statistical analyses were performed using GraphPad Prism 7.02 (GraphPad Software Inc., San Diego, CA, USA). Unpaired t-test was used to evaluate the difference between treatment and control groups. One-way ANOVA was used to evaluate the difference when three groups were included. All means are reported with SEM. *p*-values < 0.05 were considered statistically significant.

### 4.14. Data Availability

RNA-seq data have been deposited in the National Center for Biotechnology Information Gene Expression Omnibus (GEO) and are accessible through GEO Series accession number GSE132410. Differentially expressed genes in HUVECs with or without IS stimulation is presented in the Appendix A.

## Figures and Tables

**Figure 1 toxins-11-00454-f001:**
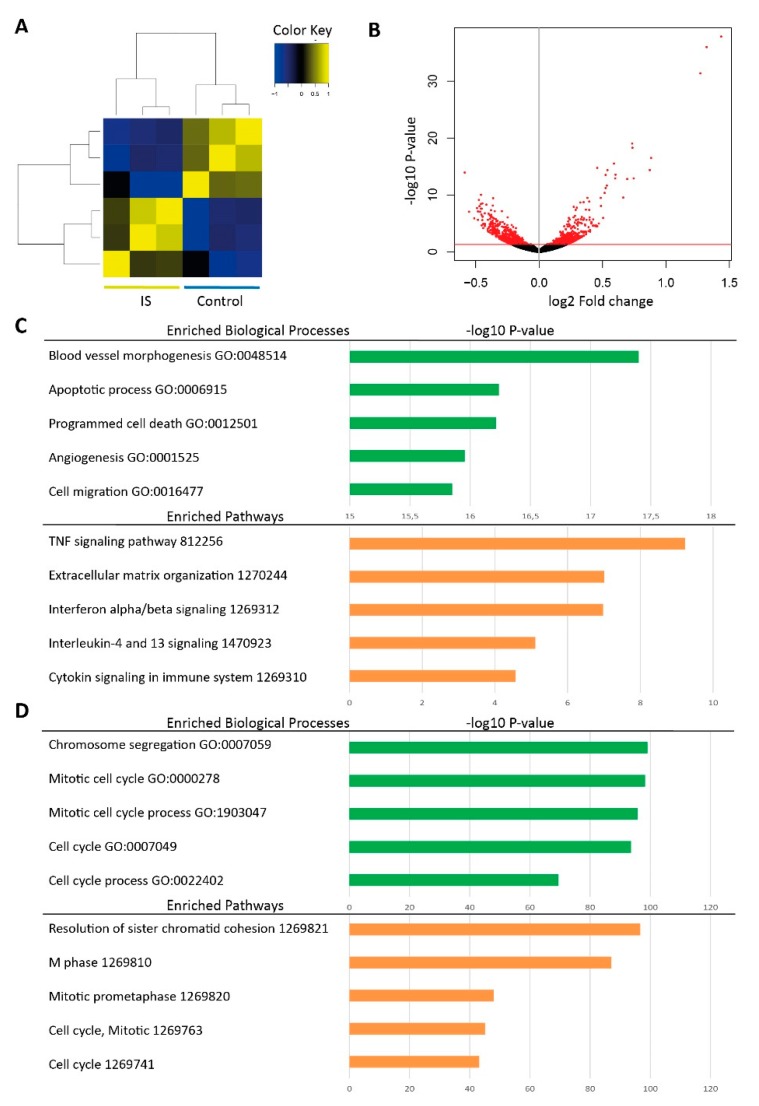
Comparison of the transcriptome profile of HUVECs treated with IS to KCl treated control groups. (**A**) Heatmap depicting clustering of samples based on all differentially expressed genes between two groups. (**B**) Volcano plot presenting fold change (*x*-axis) and *p* value (*y*-axis) of all genes in log 2 scale. Differentially expressed genes are shown in red. (**C**) Top five enriched biological processes (green) and pathways (orange) based on IS upregulated genes. (**D**) Top five enriched biological processes (green) and pathways (orange) based on IS downregulated genes.

**Figure 2 toxins-11-00454-f002:**
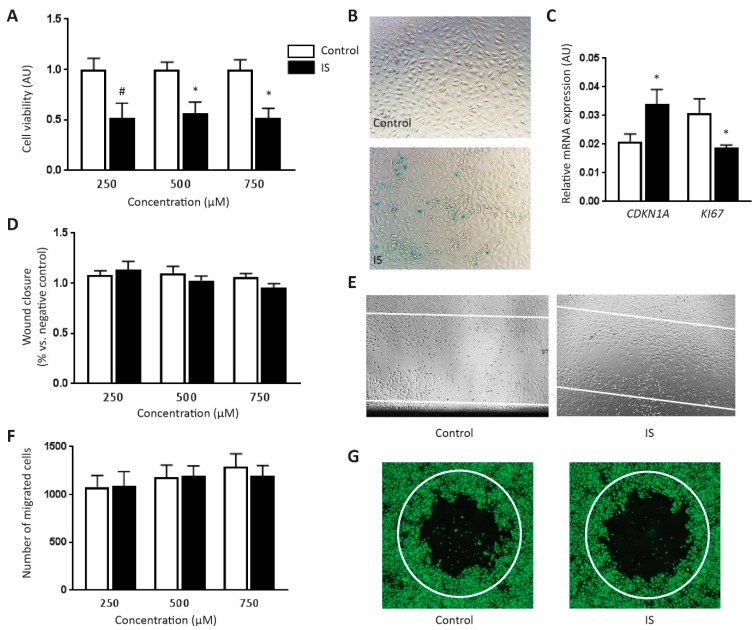
Functional examination of the identified IS related processes: Assessment of cell viability, senescence and migration in response to IS. (**A**) Cell viability was studied using a MTT assay. IS diminished viability of HUVECs when compared to the control at three difference concentrations (*n* = 4). (**B**) A representative image of X-gal activity in HUVECs treated with 250 μM IS or KCl control buffer, at 10× magnification. More X-gal positive cells (blue) were observed in IS group when compared to the control. (**C**) RT-qPCR results showed a higher expression level of cell senescence marker CDKN1A and a lower expression level of cell proliferation marker KI67 in HUVECs after exposing to 250 μM IS when compared to the KCl control (*n* ≥ 5). (**D**) An invasive wound healing assay was performed to study the influence of IS on cell migration ability. No difference on the migration distances of HUVECs was shown between two groups at three difference concentrations (*n* = 3). (**E**) Examples of migrated HUVECs after exposure to 250 μM IS or KCl control after 24 h in the wound healing assay, at 4× magnification. (**F**) A non-invasive plug assay was also performed to study the influence of IS on cell migration ability. No difference on the migrated HUVECs into the cell-free area was shown between two groups at three difference concentrations (*n* = 6). (**G**) Examples of migrated HUVECs after exposure to 250 μM IS or KCl control after 24 h in the plug assay. 2× magnification was used. All values are presented as mean ± SEM and they are shown in arbitrary units (AU), # *p* value < 0.1, * *p* value < 0.05. White lines indicate migration area in wound healing and plug assay.

**Figure 3 toxins-11-00454-f003:**
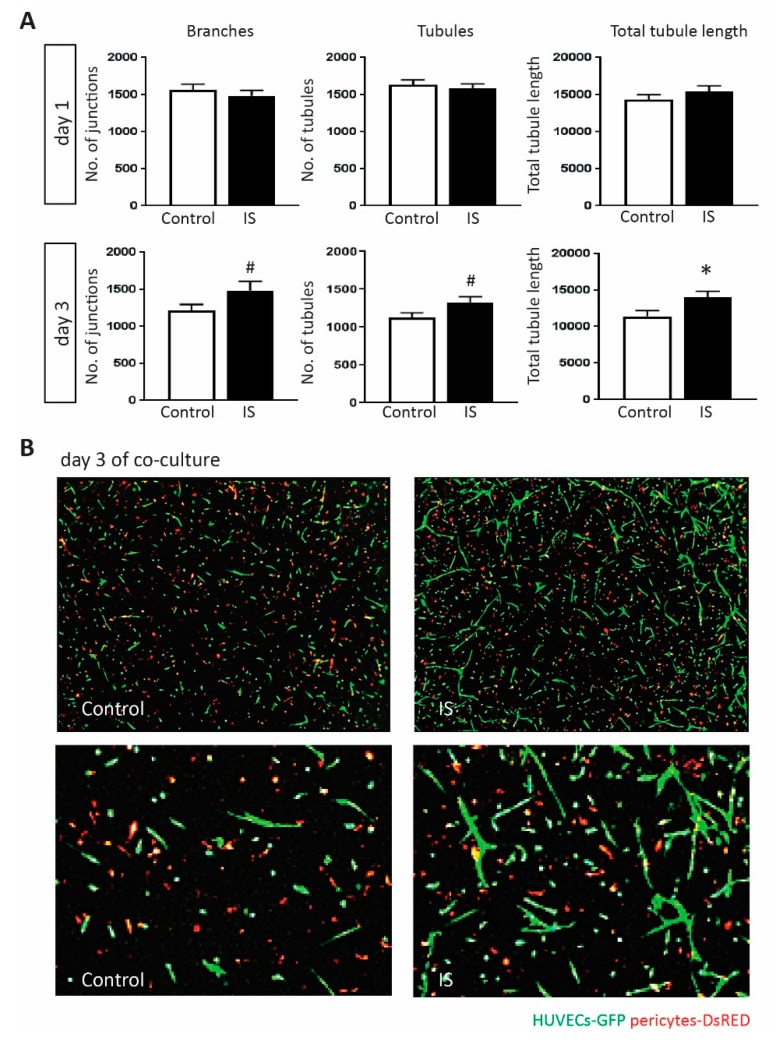
Examination of IS affected angiogenic responses. (**A**) A 3D co-culture model, in which HUVECs-GFP cells and pericytes-DsRED were cultured together in type I collagen, was used to study the influence of IS on angiogenesis at 250 μM. No difference on the number of branches, the number of tubules, and the total tubule length was detected between the two groups at 24 h after incubation. After 3 days, IS showed a tendency to promote angiogenesis by increasing all three parameters when compared to the KCl control (*n* ≥ 25). (**B**) Confocal images showing representative examples of vascular formation at day 3 post 250 μM IS or KCl control stimulation. Images shown in the upper row were taken at 20× magnification and zoomed-in views are shown in the lower row. In red are shown DsRED marked pericytes. In green are shown GFP marked HUVECs. All values are mean ± SEM, # *p* value < 0.1, * *p* value < 0.05.

**Figure 4 toxins-11-00454-f004:**
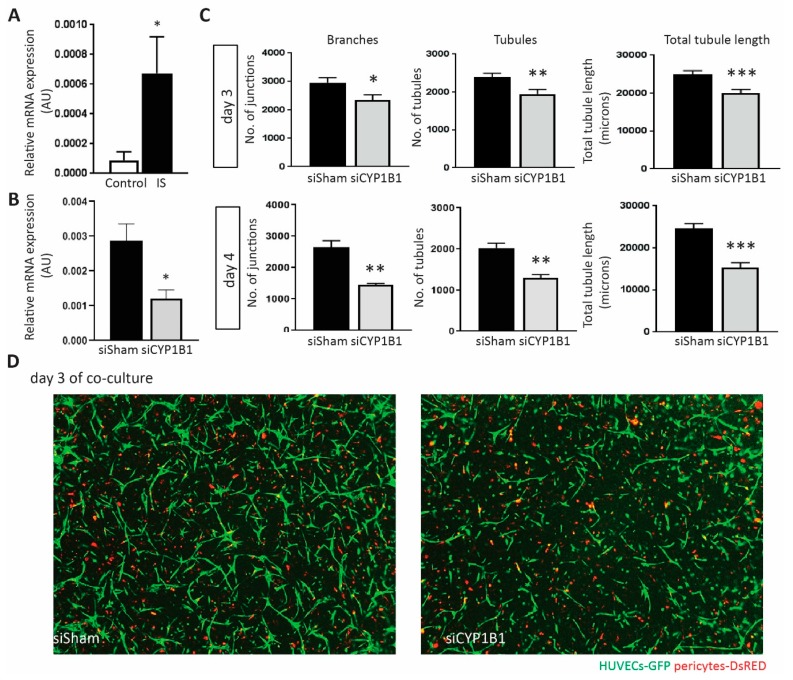
Investigation of increased *CYP1B1* expression in IS influenced angiogenic responses. (**A**) Bar graphs show results of RT-qPCR evaluation of CYP1B1 expression level in HUVECs treated with 250 μM IS compared to the KCl controls (*n* = 6). (**B**) Bar graphs show results of RT-qPCR evaluation of CYP1B1 expression level in *CYP1B1* silenced HUVECs when compared to siSham transfected cells at day 3 post transfection (*n* = 4). (**C**) Bar graphs show the number of branches, the number of tubules, and the total tubule length in *CYP1B1* targeting siRNA transfected HUVECs (siCYP1B1) compared to sham transfected HUVECs (siSham) 3 days and 4 days after 250 μM IS stimulation (*n* ≥ 3). (**D**) Examples of vascular formation at day 3 post 250 μM IS in siSham and siCYP1B1 treated HUVECs. In red are shown DsRED marked pericytes. In green are shown GFP marked HUVECs. 20× magnification was used. All values are presented as mean ± SEM and they are shown in arbitrary units (AU), * *p* value < 0.05, ** *p* value < 0.01, *** *p* value < 0.001.

**Figure 5 toxins-11-00454-f005:**
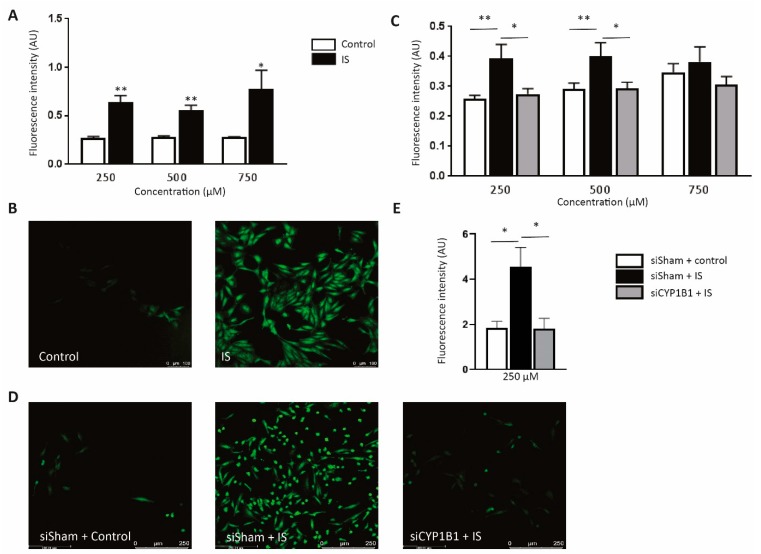
Detection of intracellular ROS production and the involvement of CYP1B1 in IS influenced ROS level. (**A**) IS or control treated HUVECs were loaded with CM-H2DCFDA, a fluorescent indicator for ROS. Bargraphs show detected fluorescent signal representing ROS production in IS group when compared to the KCl control at three difference concentrations (*n* = 4). (**B**) Confocal figures showing representative examples of ROS signals in HUVECs treated with 250 μM IS compared to the KCl controls. (**C**) ROS signal measured in HUVECs treated with siRNA targeting CYP1B1 (siCYP1B1) versus non-targeting siRNA shams (siSham) with different doses of IS or equivalent doses of KCl control stimulation. (*n* ≥ 9 for each group). (**D**) Confocal figures showing typical examples of ROS signals in siSham or siCYP1B1 treated HUVECs at 24 h after exposure to 250 μM IS or KCl control. 20× magnification was used. (**E**) ROS signal measured in CMECs treated with siCYP1B1 versus siSham (*n* = 3) with or without 250 μM IS stimulation. All values are mean ± SEM and they are shown in arbitrary units (AU), * *p* value < 0.05, ** *p* value < 0.01.

**Table 1 toxins-11-00454-t001:** Top 10 genes that were significantly up- or down-regulated in IS-treated HUVECs when compared to the KCl treated control.

Category	Ensembl ID	Gene Symbol	Gene Name	Angiogenic Function ^†^	Fold Change (log2)	*p*-Value
Up-regulation	ENSG00000138061	*CYP1B1*	Cytochrome P450 Family 1 Subfamily B Member 1	Promote angiogenesis	1.434	1.390 × 10^−38^
	ENSG00000114812	*VIPR1*	Vasoactive Intestinal Peptide Receptor 1	Not known	1.321	1.072 × 10^−36^
	ENSG00000137809	*ITGA11*	Integrin Subunit Alpha 11	Not known	1.272	4.190 × 10^−32^
	ENSG00000178695	*KCTD12*	Potassium Channel Tetramerization Domain Containing 12	Not known	0.883	3.247 × 10^−17^
	ENSG00000063438	*AHRR*	Aryl-Hydrocarbon Receptor Repressor	Not known	0.872	4.144 × 10^−15^
	ENSG00000007908	*SELE*	Selectin E	Not known	0.746	1.240 × 10^−13^
	ENSG00000137331	*IER3*	Immediate Early Response 3	Not known	0.736	5.396 × 10^−19^
	ENSG00000163659	*TIPARP*	TCDD Inducible Poly(ADP-Ribose) Polymerase	Not known	0.734	8.865 × 10^−20^
	ENSG00000144476	*ACKR3*	Atypical Chemokine Receptor 3	Promote angiogenesis	0.695	1.596 × 10^−13^
	ENSG00000144802	*NFKBIZ*	NFKB Inhibitor Zeta	Not known	0.663	3.022 × 10^−10^
Down-regulation	ENSG00000117724	*CENPF*	Centromere Protein F	Not known	−0.586	1.114 × 10^−14^
	ENSG00000143476	*DTL*	Denticleless E3 Ubiquitin Protein Ligase Homolog	Not known	−0.554	8.619 × 10^−8^
	ENSG00000163808	*KIF15*	Kinesin Family Member 15	Not known	−0.510	1.195 × 10^−6^
	ENSG00000138778	*CENPE*	Centromere Protein E	Not known	−0.490	1.912 × 10^−8^
	ENSG00000137812	*CASC5*	Kinetochore Scaffold 1	Not known	−0.483	8.198 × 10^−8^
	ENSG00000184661	*CDCA2*	Cell Division Cycle Associated 2	Not known	−0.482	2.314 × 10^−6^
	ENSG00000066279	*ASPM*	Abnormal Spindle Microtubule Assembly	Not known	−0.473	7.741 × 10^−10^
	ENSG00000156802	*ATAD2*	ATPase Family AAA Domain Containing 2	Inhibit angiogenesis	−0.470	9.314 × 10^−9^
	ENSG00000196549	*MME*	Membrane Metalloendopeptidase	Not known	−0.465	9.572 × 10^−8^
	ENSG00000132646	*PCNA*	Proliferating Cell Nuclear Antigen	Not known	−0.461	3.809 × 10^−9^

†: Established biological function of each gene in relation to angiogenesis was collected from NCBI Gene (https://www.ncbi.nlm.nih.gov/gene).

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
