# Peer review of "Indoxyl Sulfate Stimulates Angiogenesis by Regulating Reactive Oxygen Species Production via CYP1B1"

_toxins, 2019, doi:10.3390/toxins11080454_

Round 1

Reviewer 1 Report

1. what were the criteria for the IS dose selection in transcriptomic analyses?

2. - IS is a protein bound uremic toxin. How is this fact take in consideration in your study design, dose selection and model development? Include this issue in the discussion

3. please insert in the table the name and relation to EC angiogenesis already described.

4. Figure 4 insert the units in total tubule length

5. In the reviewer’s opinion, CYP1B1 western blot and activity assay is missing.

6. At the beginning of the discussion is said that this is We identified CYP1B1 as a new downstream target of IS. It is already known that cyp1b1 is a target gene of IS, or not? Is IS activating AhR canonic pathway with activation of Cyp1b1 and AhR or are there other pathways linking IS to cyp1b1. If it is AhR, wouldn´t be expected to find an increase in cyp1a1 and 1a2?

7.  change figure 2d. 2e for 2D and 2E

8. Line 91 head IS induces cell senescence would be section 2.3.

9. … carcinoma when compared to renal kidney, please correct

Reviewer 2 Report

The content of this paper is interesting. This is considered worthy to be published in this journal.

Author Response

Title: Indoxyl sulfate stimulates angiogenesis by regulating reactive oxygen species production via CYP1B1

Reviewer 2

Comments and Suggestions for Authors:

The content of this paper is interesting. This is considered worthy to be published in this journal.

We thank the reviewer for considering our paper suitable for publication in this journal.

Reviewer 3 Report

Normally, uremic toxins including indoxyl sulfate (IS) is recognized to worsen tissue and cell damage via oxidative stress and inflammation. Regarding angiogenesis, as the authors described, IS attenuates revascularization in response to ischemia through the suppression of endothelial progenitor cells in vivo and invitro experiments (Hung, et al. Kidney Int. 2016). Moreover, IS concentration is clinically associated with peripheral artery disease in patients on hemodialysis (Li, et al. Atherosclerosis. 2012). In the present study, the authors described that IS promoted tube formation nevertheless IS promoted senescence and apoptosis in HUVECs, and their conclusion disagrees previous findings of IS on angiogenesis.

1)    What is the clinical implication? It is unclear and the critical point in this study.

2)    The conclusion was contraindication with the general role of IS on angiogenesis. How did the authors think the conflicting results?  The present results might simply show acute effect of ROS induced by IS on angiogenesis in in vitro experiment. IS enhanced cell senescence and apoptosis, causing the impaired angiogenesis. Therefore, the present data is not convincing at all.m In vivo animal study should be performed to clarify IS action on angiogenesis (e.g. Hindlimb ischemia model with or without direct IS administration)

3)    For the in vitro studies, they used EGM2 and Dulbecco which did not contain albumin. This is a major quality problem for studies on protein bound uremic toxins (for explanation, see Vanholder et al, JASN 2014). For biological effects to be relevant, they should be assessed at free concentrations as they occur in uremia. The concentrations shown are far above that (are even above the high range total concentrations in uremia, see Duranton et al, JASN, 2012). That concentration in culture medium without albumin is above the high uremic range and not representative for most CKD patients.

4)    IS is well-known to exert its biological action via aryl hydrocarbon receptor (AhR)-mediated pathway. AhR ubiquitously exists in various tissue and cells including endothelial cells. How about the involvement of AhR on IS-induced tube formation ?

5)    In discussion, the paragraph seems not to be needed (line 271-281).

Round 2

Reviewer 1 Report

Pag 7 line 157 - figure 4B and not 5B

Reviewer 3 Report

Several concerns are resolved by the author’ responses, however, the clinical implication still remains unclear.

The authors claimed that IS can induce early angiogenic response at initial phase, and IS-induced cell senescence and apoptosis would eventually lead to the lack of further vascular expansion. However, they present no data to support the above speculation. An additional experiment was necessary for clarify it. For example, IS-induced angiogenesis was seen in initial phase (day3). If it would be observed for long time (more than 3 days observation), further vascular expansion were lacking treated with IS compared to control ?

Round 3

Reviewer 3 Report

There is no further comment.